# Natural Product Isoliquiritigenin Activates GABA_B_ Receptors to Decrease Voltage-Gate Ca^2+^ Channels and Glutamate Release in Rat Cerebrocortical Nerve Terminals

**DOI:** 10.3390/biom11101537

**Published:** 2021-10-18

**Authors:** Tzu-Yu Lin, Cheng-Wei Lu, Pei-Wen Hsieh, Kuan-Ming Chiu, Ming-Yi Lee, Su-Jane Wang

**Affiliations:** 1Department of Anesthesiology, Far-Eastern Memorial Hospital, Pan-Chiao District, New Taipei City 22060, Taiwan; drlin1971@gmail.com (T.-Y.L.); drluchengwei@gmail.com (C.-W.L.); 2Department of Mechanical Engineering, Yuan Ze University, Taoyuan 32003, Taiwan; 3Research Center for Chinese Herbal Medicine, College of Human Ecology, Chang Gung University of Science and Technology, Taoyuan 33303, Taiwan; pewehs@mail.cgu.edu.tw; 4Graduate Institute of Natural Products, School of Traditional Chinese Medicine, and Graduate Institute of Biomedical Sciences, College of Medicine, Chang Gung University, Taoyuan 33303, Taiwan; 5Department of Anesthesiology, Chang Gung Memorial Hospital, Taoyuan 33305, Taiwan; 6Division of Cardiovascular Surgery, Cardiovascular Center, Far-Eastern Memorial Hospital, New Taipei 22060, Taiwan; chiu9101018@gmail.com (K.-M.C.); mingyi.lee@gmail.com (M.-Y.L.); 7Department of Nursing, Asia Eastern University of Science and Technology, New Taipei City 22060, Taiwan; 8Department of Photonics Engineering, Yuan Ze University, Taoyuan 32003, Taiwan; 9School of Medicine, Fu Jen Catholic University, New Taipei City 24205, Taiwan

**Keywords:** isoliquiritigenin, GABA_B_ receptors, G_βγ_, VGCCs, glutamate, cerebrocortical synaptosomes

## Abstract

Reduction in glutamate release is a key mechanism for neuroprotection and we investigated the effect of isoliquiritigenin (ISL), an active ingredient of Glycyrrhiza with neuroprotective activities, on glutamate release in rat cerebrocortical nerve terminals (synaptosomes). ISL produced a concentration-dependent inhibition of glutamate release and reduced the intraterminal [Ca^2+^] increase. The inhibition of glutamate release by ISL was prevented after removing extracellular Ca^2+^ or blocking P/Q-type Ca^2+^ channels. This inhibition was mediated through the γ-aminobutyric acid type B (GABA_B_) receptors because ISL was unable to inhibit glutamate release in the presence of baclofen (an GABA_B_ agonist) or CGP3548 (an GABA_B_ antagonist) and docking data revealed that ISL interacted with GABA_B_ receptors. Furthermore, the ISL inhibition of glutamate release was abolished through the inhibition of G_i/o_-mediated responses or G_βγ_ subunits, but not by 8-bromoadenosine 3′,5′-cyclic monophosphate or adenylate cyclase inhibition. The ISL inhibition of glutamate release was also abolished through the inhibition of protein kinase C (PKC), and ISL decreased the phosphorylation of PKC. Thus, we inferred that ISL, through GABA_B_ receptor activation and G_βγ_-coupled inhibition of P/Q-type Ca^2+^ channels, suppressed the PKC phosphorylation to cause a decrease in evoked glutamate release at rat cerebrocortical nerve terminals.

## 1. Introduction

Isoliquiritigenin (ISL, Figure 1A), a flavonoid with a chalcone structure, is the main active ingredient in the *Glycyrrhiza glabra* L. root. ISL has received attention because of its various pharmacological benefits, including antibacterial, antiviral, antidiabetic, antioxidant, anti-inflammatory, anticarcinogenic, analgesic, and neuroprotective effects [1,2,3]. Regarding its neuroprotective activity, previous animal studies found that ISL can penetrate the blood-brain barrier [4] and protects against brain damage and cognitive impairment induced by kainic acid, lipopolysaccharide (LPS), ischemia, or traumatic brain injury [5,6,7]. In vitro, ISL attenuates glutamate-, H_2_O_2_- or amyloid beta-protein (Aβ) (25–35)-induced neuronal cell death by reducing the cellular Ca^2+^ concentration and reactive oxygen species (ROS) levels in HT22 and PC12 cells, as well as in the cultured cortical neurons of rats [8,9,10]. Although the mechanisms underlying the neuroprotective effects of ISL are not fully clarified, it has been reported that this beneficial effect is related to its anti-inflammatory and antioxidant activities [5,8,11,12].

Glutamate is the predominant excitatory neurotransmitter in the central nervous system (CNS) that produces excitation through glutamate receptors, supporting normal synaptic transmission and sustaining learning and memory processes [13,14]. However, in various pathological conditions, including stroke and neurodegenerative diseases, excessive glutamate release may mediate neuronal injury or death [15,16]. Thus, reduced glutamatergic transmission through the inhibition of released glutamate has been proposed as a neuroprotective mechanism [17,18,19]. Studies have reported that several natural products with neuroprotective effects may reduce presynaptic glutamate release [20,21,22,23]. Likewise, ISL has a neuroprotective-like effect and whether or not ISL can influence presynaptic glutamate release should be evaluated. Therefore, this study investigated the effect and possible mechanisms of ISL on glutamate release in rat cerebrocortical nerve terminals (synaptosomes).

## 2. Materials and Methods

### 2.1. Drugs

Isoliquiritigenin (ISL; purity > 98%) was purchased from ChemFaces (Wuhan, Hubei, China). 4-aminopyridine (4-AP), baclofen, (3-Aminopropyl)(diethoxymethyl)phosphinic acid (CGP35348), bafilomycin A1, isindolylmaleimide I (GF109203X), 5,6,7,13-tetrahydro-13-methyl-5-oxo-12*H*-indolo[2,3-*a*]pyrrolo[3,4-*c*]carbazole-12-propanenitrile (Go6976), *cis*-N-(2-Phenylcyclopentyl)-azacyclotridec-1-en-2-amine hydrochloride (MDL12330A), gallein, and nifedipine were purchased from Tocris Cookson (Bristol, UK). Fura-2-acetoxymethyl ester (Fura-2-AM) was purchased from Invitrogen (Carlsbad, CA, USA). ω-conotoxin GVIA and ω-agatoxin IVA were purchased from Alomone lab (Jerusalem, Israel). PTX, 8-bromo-cAMP and all other reagents were purchased from Sigma-Aldrich (St. Louis, MO, USA).

### 2.2. Animals

Male Sprague-Dawley rats (150–200 g) were maintained at 21–23 °C with free access to water and food, under a 12:12 h light/dark cycle (lights-on at 07:00 h). The procedures used in this study were performed in accordance with the National Institutes of Health Guide for the Care and Use of Laboratory Animals. The experiments were developed after the protocol approved by the Institutional Ethics Committee (A10814). In this study, all efforts were made to minimize the number of animals used and their suffering.

### 2.3. Preparation of Synaptosomes

Rats were killed by cervical dislocation and decapitation. The cerebral cortex was rapidly removed and homogenized in ice-cold hepes-buffered medium containing 0.32 M sucrose (pH 7.4). The homogenate was centrifuged at 3000× *g* for 10 min at 4 °C. The supernatant was retained and centrifuged at 14,500× *g* for 12 min at 4 °C. The pellet was resuspended and layered on top of a discontinuous Percoll gradient and centrifuged at 32,500× *g* for 7 min at 4 °C. Protein concentration was determined using the Bradford assay. Synaptosomes were centrifuged in the final wash to obtain synaptosomal pellets with 0.5 mg protein [24,25].

### 2.4. Glutamate Release

Synaptosomal pellets were analyzed for glutamate release using enzyme-linked fluorescence method previously described [24,26]. As synaotosome is not amenable to electrical stimulation, a number of biochemical depolarization protocols have been developed, including the use of K^+^ channel blocker 4-AP or high external [K^+^] [24]. Therefore, synaptosomes were suspended in hepes-buffered medium containing 2 mM NADP^+^, 50 units of glutamate dehydrogenase, and 1.2 mM CaCl_2_, and the synaptosome suspension was stimulated with either 1 mM 4-AP or 15 mM KCl. Increases in fluorescence due to production of NADPH was determined using a PerkinElmer LS55 spectrofluorimeter with excitation and emission wavelengths set at 340 nm and 460 nm, respectively. Released glutamate was calibrated by a standard of exogenous glutamate (5 nmol) and expressed as nanomoles glutamate per milligram synaptosomal protein (nmol/mg protein). Values quoted in the text and depicted in bar graphs represent the levels of glutamate cumulatively released after 5 min of depolarization, and are expressed as nmol/mg protein/5 min.

### 2.5. Cytosolic Free Ca^2+^ Concentration ([Ca^2+^]_C_)

Synaptosomes were incubated in the hepes-buffered medium containing 5 µM Fura-2 AM and 0.1 mM CaCl_2_ for 30 min at 37 °C. After Fura-2 loading, synaptosomes were centrifuged for 1 min at 3000× *g* and pellets were resuspended in hepes-buffered medium containing 1.2 mM CaCl_2_. Fura-2/Ca fluorescence was monitored in a Perkin-Elmer LS55 spectrofluorimeter at excitation wavelengths of 340 nm and 380 nm (emission wavelength 505 nm), 340/380 fluorescence ratios (F) were calculated. [Ca^2+^]_C_ (nM) was calculated following calibration procedures [27], using 0.1% sodium dodecyl sulfate to obtain the maximal fluorescence (F_max_) with Fura-2 saturation with Ca^2^^+^, followed by 10 mM EGTA (Tris buffered) to obtain minimum fluorescence (F_min_) in the absence of any Fura-2/Ca^2^^+^ complex. [Ca^2+^]_C_ (nM) was calculated by the equation ([Ca^2+^] = *K_d_* (F − F_min_/F_max_ − F)) [28], using a *K*_d_ of 210 nM for the Fura-2/Ca^2+^ complex.

### 2.6. Immunohistochemistry

Immunofluorescence analysis was performed on synaptosomes as described previously [29,30]. The synaptosomes were attached to the polylysine-coated coverslips for 40 min, fixed with 4% paraformaldehyde for 5 min, and permeabilized with 0.2% Triton X-100 in phosphate-buffered saline (PBS) for 1 h. Subsequently, the synaptosomes were incubated with primary antibody solutions containing anti-GABA_B_ receptor antibody (1:100; Abcam, Cambridge, UK) and anti-vesicular glutamate transporter 1 (VGLUT1) antibody (1:100; Abcam, Cambridge, UK) overnight. Synaptosomes were then washed with PBS and incubated in a mixture of goat anti-mouse DyLight 549- and goat anti-rabbit fluorescein isothiocyanate (FITC)-conjugated secondary antibodies (1:200; Invitrogen, Carlsbad, CA, USA) for 2 h at room temperature. Immunoreactivity was visualized on a Image Xpress Micro confocal microscope (Molecular Devices, San Jose, CA, USA). The estimation of the percentage of glutamatergic terminals positive for GABA_B_ receptor was counted three randomly selected areas (255 × 255 µm^2^) from each coverslip and averaged using ImageJ (Bio-Rad, Hercules, CA, USA).

### 2.7. Molecular Docking Study

The tools of CDOCKER in software of Discovery Studio 4.1 client was used for the molecular docking experiment. Firstly, the molecular structure of GABA_B_ protein (PDB ID 4MGF) was downloaded from the RCSB Protein Data Bank, and then prepared by the molecular modeling software. The structure of ISL was established following the standard protocols in software, and then docked into the binding site of the active site in GABA_B_ protein.

### 2.8. Western Blotting

Synaptosomes were lysed in an ice-cold Tris-HCl buffer solution and then centrifuged for 10 min at 13,000× *g* at 4 °C. Protein content was determined by using the Bradford assay. Equal amounts (30 µg) of samples were loaded per lane onto 10% polyacrylamide gel, and then transferred to a polyvinylidenedifluoride (PVDF) membrane in a semi-dry system (Bio-Rad, Hercules, CA, USA) for 120 min at a constant current of 0.15 mA. Membranes were blocked with Tris-buffered solution that contained 4% bovine serum albumin for 1 h at room temperature under agitation. After blocking, membranes were incubated overnight at 4 °C with the primary antibodies (anti-protein kinase C (PKC), 1:700; anti-phospho-PKC, 1:2000; anti-protein kinase C alpha (PKCα), 1:600; anti-phospho-PKCα, 1:2000; anti-phospho-MARCKS, 1:250; anti-β-actin, 1:1000; Cell Signaling Technology, Beverly, MA, USA). The immunoreactive bands were visualized by using peroxidase-conjugated donkey anti-rabbit IgG secondary antibodies (1:1000; Cell Signaling Technology, Beverly, MA, USA) and enhanced chemiluminescence (ECL; Amersham, Buckinghamshire, UK). After protein detection, densitometric analyses were performed using ImageJ software.

### 2.9. Statistical Analysis

The data were checked for normal distribution and analyzed by Student *t*-tests or one-way analysis of variance (ANOVA), followed by Fisher’s LSD multiple comparisons tests as appropriate using GraphPad Prism 7 (La Jolla, CA, USA). Data are expressed as mean values ± stander error of the mean (SEM) of at least five independent experiments. A *p*-value < 0.05 was considered statistically significant.

## 3. Results

### 3.1. ISL Inhibits 4-AP-evoked Glutamate Release from Nerve Terminals in the Cerebral Cortex through Ca^2+^ Influx Reduction

To examine the presynaptic action of ISL, isolated synaptosomes were depolarized with 4-APwhich has been shown to open voltage-gated Ca^2^^+^ channels (VGCCs) and to induce the release of glutamate [31]. Under control conditions, 4-AP evoked a glutamate release of 7.2 ± 0.1 nmol/mg/5 min. Incubation with 10 µM ISL for 10 min prior to the addition of 1 mM 4-AP produced an inhibition of 4-AP-evoked glutamate release to 4.3 ± 0.1 nmol/mg/5 min [t(8) = 61.5, *p* < 0.001]. The inhibition of 4-AP-evoked glutamate release by ISL was concentration-dependent, with a maximum inhibition of 78.6 ± 0.5% produced at 50 µM; the half maximal effective concentration for this inhibition was 17.2 µM (Figure 1B). Given the robust repression of evoked glutamate release that was seen with 10 µM ISL, this concentration of ISL was used in subsequent experiments to evaluate the mechanisms that underlie the ability of ISL to reduce glutamate release.

To investigatewhether the inhibition of 4-AP-evoked glutamate release by ISL was mediated by an effect on exocytotic vesicular release, we examine the action of ISL in the presence of bafilomycin A1 (0.1 µM), which causes the depletion of glutamate in synaptic vesicles. As shown in Figure 1C, bafilomycin A1 reduced control 4-AP-evoked glutamate release (31.8 ± 2.7% of control, *p* < 0.001). When bafilomycin A1 was present, 10 µM ISL failed to produce significant inhibition (31.2 ± 2.1% of control, *p* = 0.98 vs. bafilomycin A1-treated group). Similarly, the glutamate release evoked by 1 mM 4-AP was reduced in the presence of extracellular-Ca^2+^-free solution that contained 300 µM EGTA (26.4 ± 1.5% of control, *p* < 0.001 vs. control group). This Ca^2+^-independent release evoked by 4-AP was unaffected by 10 µM ISL (26.8 ± 1.5% of control, *p* = 0.99; Figure 1D). In addition, we examined the effect of ISL on the 15 mM KCl-evoked glutamate release, a process that involves Ca^2+^ influx primarily through the opening of VGCCs [32]. As illustrated in Figure 1E, 10 µM ISL significantly inhibited the KCl-evoked glutamate release [60.4 ± 6.7% of control, t(8) = 5.9, *p* < 0.001 vs. control]. These results indicate that the observed inhibition of glutamate release by ISL is likely to be due to a decrease in Ca^2+^ influx through VGCCs which are coupled to glutamate exocytosis in the nerve terminals. To verify this, [Ca^2+^]_C_ was determined in synaptosomes preloaded with Fura-2. Stimulation of synaptosomes with 1 mM of 4-AP elicited a rise in [Ca^2+^]_C_. 10 µM ISL preincubation did not significantly affect basal [Ca^2+^]_C_ (control, 148.6 ± 1.1; ISL, 147.7 ± 2.1; *p* = 1 vs. control group; Figure 1F), but caused a significant decrease in the 4-AP-induced [Ca^2+^]_C_ elevation [64.1 ± 3.7% of control, t(8) = 50.9, *p* < 0.001 vs. control group].

### 3.2. Specifics of VGCCs Involved in ISL Inhibition of Glutamate Release

In the rat cerebrocortical nerve terminal preparation, the release of glutamate is primarily coupled to the entry of Ca^2+^ through both N- and P/Q-type Ca^2+^ channels (39,40). The results above described indicated that the ISL-induced inhibition of glutamate release involves a reduction in Ca^2+^ influx through VGCCs. To verify this hypothesis, we used selective VGCC inhibitors to characterize the role of individual Ca^2+^ channel subtypes in the observed ISL-mediated inhibition of glutamate release. As shown in Figure 2A, with the blockade of N-type VGCCs with ω-conotoxin GVIA (2 µM), glutamate release evoked by 4-AP under control conditions was reduced (70.6 ± 1.2% of control, *p* < 0.001). When ω-conotoxin GVIA was present, glutamate release was further inhibited by 10 μM ISL (39.6 ± 0.7% of control, *p* < 0.05 vs. ω-conotoxin GVIA -treated group, Figure 2A). The additive relation between ω-conotoxin GVIA and ISL indicates that N-type VGCCs appear not to mediate the action of ISL on glutamate release. In addition, selective blockade of P/Q-type VGCCs using ω-agatoxin IVA (0.5 µM) reduced control 4-AP-evoked glutamate release (44.7 ± 0.6% of control, *p* < 0.001). Notably, in the presence of ω-agatoxin IVA, 10 µM ISL inhibition of glutamate release was completely abolished (45.4 ± 0.4% of control, *p* = 0.92 vs. ω-agatoxin IVA-treated group, Figure 2B), indicating that P/Q-type VGCCs are involved in the observed modulation of glutamate release by ISL. In contrast, selective blockade of L-type VGCCs with nifedipine (1 µM) caused no significant changes on glutamate release evoked by 4-AP under control conditions (98.6 ± 1.5% of control, *p* = 0.83), indicating that L-type VGCCs are not directly support glutamate release from synaptosomes. When nifedipine was present, 10 µM ISL significantly reduced 4-AP-evoked glutamate release [42.4 ± 1.4% of control, *p* < 0.05 vs. nifedipine-treated group, Figure 2C), indicating that L-type VGCCs, similar to N-type VGCCs, are not involved. These results demonstrate that a reduction in Ca^2+^ influx mediated by P/Q-type VGCCs is associated with the inhibition of glutamate release by ISL.

### 3.3. Protein Kinase C Suppression Is Involved in the ISL-mediated Inhibition of Glutamate Release

It has been reported that increased Ca^2+^ influx in nerve terminals enhances PKC activation and glutamate release [33]. We therefore examined whether the inhibition of Ca^2+^ influx caused by ISL decreased PKC activity. We tested how GF109203X, a general inhibitor of PKC, changed the ISL effect. Specifically, in the absence of GF109203X, ISL significantly reduced the 4-AP-evoked glutamate release (*p* < 0.001). Application of 10 µM GF109203X significantly inhibited control 4-AP-evoked glutamate release (61.1 ± 0.9% of control, *p* < 0.001). In the presence of 10 µM GF109203X, 10 µM ISL failed to reduce 4-AP-evoked glutamate release (60.8 ± 0.7% of control, *p* = 0.98 vs. GF109203X-treated group, Figure 3A). Similar results were also observed in the presence of 10 µM of Go6976, an inhibitor of conventional Ca^2+^-dependent PKCα isozymes, which reduced the 4-AP-evoked glutamate release (69.4 ± 1.3% of control, *p* < 0.001). In addition, 10 µM ISL did not reduce the release of glutamate evoked by 4-AP when Go6976 was present (67.6 ± 0.9% of control, *p* = 0.49 vs. Go6976-treated group, Figure 3B). To verify that the PKC suppression was involved in ISL inhibition of glutamate release, we examined the effect of ISL on the phosphorylation of PKC and its substrate, myristoylated alanine-rich C kinase substrate (MARCKS), in cerebrocortical synaptosomes. Western blot analysis showed that 1 mM of 4-AP increased the phosphorylation of PKC (892.9 ± 42.5% of control, *p* < 0.001), PKCα (437.3 ± 81.5% of control, *p* < 0.01), and MARCKS (136.8 ± 91.6% of control, *p* < 0.001) in the presence of 1.2 mM CaCl_2_. These phosphorylation changes caused by 4-AP were markedly attenuated by the presence of 10 µM ISL (PKC, 109.4 ± 2.3% of control; PKCα, 99.9 ± 4.8% of control; MARCKS, 91.6 ± 20.8% of control; *p* < 0.05 vs. 4-AP-treated group; Figure 3C). These results suggest that the PKC signaling pathway was suppressed by ISL during its inhibition of 4-AP-evoked glutamate release.

### 3.4. Presynaptic GABA_B_ Receptors Mediate ISL’s Effect on Glutamate Release

ISL-mediated actions in the CNS have previously been associated with the modulation of γ-aminobutyric acid type B (GABA_B_) receptors [34]. Since GABA_B_ receptor activation inhibits Ca^2+^ influx and glutamate release [35,36], we investigated whether ISL inhibition of glutamate release was dependent on GABA_B_ receptor activity. In these experiments, we first identified the presence of the GABA_B_ receptor protein in the cerebrocortical glutamatergic synaptosomes through immunocytochemical analysis. As detailed in Figure 4A, a significant percentage (71%) of vesicular transporter of glutamate type 1 (VGLUT1)-positive glutamatergic particles (red) was also immunopositive for the GABA_B_ receptor protein (green), indicated by the merged staining (yellow). In addition, the effect of the GABA_B_ receptor selective agonist baclofen on the ISL-mediated inhibition of glutamate release was evaluated. As detailed in Figure 4B, baclofen (50 µM) reduced the 4-AP-evoked release of glutamate from the cerebrocortical synaptosomes (41.2 ± 0.9% of control, *p* < 0.001). In the presence of baclofen, the addition of 10 µM ISL failed to reduce 4-AP-evoked glutamate release (40.4 ± 1.9% of control, *p* = 0.99 vs. baclofen-treated group). The lack of additivity in the inhibitory actions of ISL and baclofen on glutamate release cans be explained by the inhibition of the same action mechanism by both compounds. Similar to lack of effect of ISL in the presence of baclofen, ISL had no effect on the 4-AP-evoked glutamate release in the presence of the GABA_B_ receptor antagonist CGP35348 (100 μM) (96.4 ± 1.7% of control, *p* = 0.98 vs. CGP35348-treated group). CGP35348 alone elicited no significant effect on the 4-AP-evoked glutamate release (96.8 ± 1.2% of control, *p* = 0.39; Figure 4C). Therefore, these results suggest that GABA_B_ receptor activation is involved in the ISL-mediated inhibition of 4-AP-evoked release.

### 3.5. ISL Interacts with the GABA_B_ Receptors

In the last decade, machine learning applications is appropriate to use in pharmacological research. Among them, molecular docking is appropriate to use pharmacology to predict the target of natural products [37].To estimate the interaction between ISL and GABA_B_ receptors, a molecular docking experiment was performed using the CDOCKER tools in the Discovery Studio 4.1 client. The molecular structure of a GABA_B_ receptor (PDB ID 4MGF) was downloaded from the Research Collaboratory for Structural Bioinformatics Protein Data Bank (PDB ID 4MGF). The results revealed that ISL had hydrogen-bonding interactions with the amino acid residue Glu349. Furthermore, the residues His170 and Cys129 in LB1 domain make a van der Waal and a lipophilic interaction with isoliquiritigenin, respectively (Figure 5). Both His170 and Glu 349 are major residues of LB1 domain in GABA_B_ receptor contact to agonists [38,39,40].

### 3.6. G_βγ_-coupling Mechanism Is Involved in the ISL-mediated Inhibition of Glutamate Release

GABA_B_ receptors are G_i/o_-protein-coupled receptors that activate different downstream effectors, such as the well-characterized G_i/o_-cyclicmonophosphate (cAMP)—protein kinase A pathway [41]. To determine whether the G_i/o_ proteins were involved in the ISL-mediated inhibition of glutamate release, synaptosomes were incubated for 4 h with the Gi/o protein inhibitor pertussis toxin (PTX; 2 µg/mL) [42]. As presented in Figure 6A, PTX did not change the glutamate release evoked by 4-AP under control conditions (101.2 ± 0.8% of control, *p* = 0.98). In the presence of PTX, the addition of 10 µM ISL failed to reduce 4-AP-evoked glutamate release (99.8 ± 0.4% of control, *p* = 0.87 vs. PTX-treated group). To further elucidate the transduction pathways lying downstream of the ISL-mediated activation of G_i/o_, we tested the influence of exogenous cAMP on the effect of ISL. In the presence of the membrane-permeable cAMP analogue 8-bromo-adenosine-3′,5′-cyclic monophosphate (8-bromo-cAMP; 250 µM), control glutamate release evoked by 4-AP was increased (143.2 ± 2,5% of control, *p* < 0.001), but 10 µM ISL significantly inhibited glutamate release (87.6 ± 0.6% of control, *p* < 0.05 vs. 8-bromo-cAMP-treated group, Figure 6B). In addition, the adenylate cyclase inhibitor MDL12330A (10 µM) reduced the control glutamate release evoked by 4-AP (84.6 ± 0.9% of control, *p* < 0.001). The inhibition caused by 10 µM ISL on glutamate release was unchanged by the presence of MDL12330A (60.8 ± 2.4% of control, *p* < 0.05 vs. MDL12330A-treated group, Figure 6C). Notably, the G protein βγ (G_βγ_) subunit inhibitor gallein (10 µM) had no significant effect on the control 4-Ap-evoked glutamate release (99.4 ± 1.9% of control, *p* = 0.98); in the presence of gallein, 10 µM ISL exhibited no further effect on 4-AP-evoked glutamate release (98.6 ± 2.1% of control, *p* = 0.94 vs. gallein-treated group, Figure 6D). These results suggest that ISL-mediated action on glutamate release is dependent on G_βγ_ subunits, but not on changes in cAMP.

## 4. Discussion

ISL is an active ingredient of Glycyrrhiza, with pharmacological functions such as antioxidation, anti-inflammation, antinociception, neuroprotection, and antitumor effects [2]. To date, no data documenting the effect of ISL on glutamate release was available. Several reports have indicated that natural products may effectively reduce the release of glutamate [20,21,22,23]. We examined whether ISL exhibited an effect on glutamate release in rat cerebrocortical nerve terminals. The main study findings were as follows: (i) ISL exerts an inhibitory effect on VGCC-dependent glutamate release from cerebrocortical nerve terminals and (ii) this inhibitory action is dependent on the presynaptic GABA_B_ receptor activation and PKC suppression.

The results suggested that the ILS inhibition of glutamate release from cerebrocortical nerve terminals was associated with a decrease in Ca^2+^ influx through the VGCCs, because ISL was unable to inhibit 4-AP-evoked glutamate release in the absence of extracellular Ca^2+^ or blocking vesicular glutamate transporters and ISL reduced intraterminal [Ca^2+^]. In addition, ISL also inhibited the release of glutamate induced by 15 mM of external KCl, which is a VGCC-dependent process [31,32]. Moreover, this research demonstrated that after the blockage of P/Q-type Ca^2+^ channels, the remaining 4-AP-evoked glutamate release was unresponsive to ISL. Thus, the suppression of Ca^2+^ influx through P/Q-type Ca^2+^ channels is involved in the inhibition of glutamate release caused by ISL. P/Q-type Ca^2+^ channels have been reported to participate in triggering glutamate release from nerve terminals [43,44,45]. In the nerve terminals, VGCCs can be mediated by the membrane potential; for example, the inhibition of Na^+^ channels and Na^+^ influx or activation of K^+^ channels and K^+^ efflux leads to cell membrane hyperpolarization. This changing membrane potential closes the VGCCs, which in turn reduces [Ca^2+^]_C_ and neurotransmitter release [31,46]. Although ISL has been shown to modulate Na^+^ and K^+^ channels [3,47,48], the inhibitory effect of ISL on the VGCCs observed in our study is not caused by a change in synaptosomal membrane potential because ISL inhibited the release of glutamate evoked by 4-AP and KCl. Both of these depolarizing treatments are thought to activate VDCCs coupled to glutamate release similarly, and this should be indicated by qualitatively similar modulation if it occurs at the level of the voltage-dependent Ca^2+^ channel. The two depolarizing paradigms differ in that 4-AP-evoked glutamate release involves the action of Na^+^ andCa^2+^ channels, whereas 15 mM external KCl-evoked glutamate release involves only Ca^2+^ channels [30,31]. This indicates that Na^+^ channels are not involved in the effect of ISL on glutamate release. Furthermore, ISL did not affect the 4-AP-evokedCa^2+^-independent glutamate release, a component of glutamate release that is exclusively dependent on membrane potential [49,50]. Therefore, our findings indicate that the inhibition of release-coupled VGCCs by ISL reflects a direct effect on VGCC function.

Some ISL-mediated action in the nervous system has been associated with the modulation of GABA_B_ receptors [34]. The GABA_B_ receptor, which is a G-protein-coupled receptor, regulates a variety of intracellular signaling systems, acting on both the pre- and postsynaptic membrane [51]. Presynaptic GABA_B_ receptors can be coupled with several intracellular effector systems to inhibit glutamate release, including the inhibition of adenylate cyclase activity, membrane-delimited G_i/o_-coupled inhibition of VGCCs, and activation of G_i/o_-coupled K^+^ channels [52,53,54]. In this study, we determined that the GABA_B_ receptor protein was co-expressed with the glutamatergic terminal marker protein VGLUT1 within the same nerve terminals, indicating the presence of GABA_B_ receptors on cerebrocortical glutamatergic terminals. The activation of GABA_B_ receptors at cerebrocortical nerve terminals was involved in the ISL-mediated inhibition of glutamate release, because ISL was unable to inhibit glutamate release in the presence of the GABA_B_ receptor agonist baclofen and antagonist CGP35348. Our docking data also revealed that ISL can interact with the amino acid residues (Glu349 and His170) of the GABA_B_ receptor; both His170 and Glu 349 are major residues of the LB1 domain in GABA_B_ receptor agonist contact [38,39,40]. In addition, the ISL-mediated action through the GABA_B_ receptor may be mediated by the sequential activation of G_i/o_ proteins because the ISL inhibition of glutamate release was abolished through the inhibition of G_i/o_-mediated responses with PTX. Moreover, ISL-mediated action on glutamate release is dependent on G_βγ_ subunits, but not on changes in cAMP, as the inhibition of glutamate release caused by ISL was antagonized through galleon-induced G_βγ_ subunit inhibition; no such changes were observed with cAMP analogue 8-bromo-cAMP or with the inhibition of adenylate cyclase with MDL12330A. These data, together with the blockade of the ISL-mediated inhibition of glutamate release in the presence of the selective inhibitor of P/Q-type Ca^2+^ channels, indicated that the ISL-mediated inhibition of glutamate release was mediated by GABA_B_ receptor activation through the G_βγ_-coupled inhibition of VGCCs. G_βγ_ has been demonstrated to regulate neurotransmitter release through a direct inhibition of VGCCs [55,56], supporting our results.

The ISL inhibition of glutamate release observed in this study was associated with a suppression of the PKC pathway, because ISL failed to inhibit glutamate release in the presence of the selective PKC inhibitors GF109203X and Go6976; ISL inhibited the phosphorylation of PKC and its substrate MARCKS. PKC is a modulator of the exocytotic pathway, where it enhances both the priming and fusion steps of vesicle exocytosis through the phosphorylation of several proteins (including MARCKS) of the exocytotic machinery [57,58,59]. Increased [Ca^2+^] in nerve terminals can activate PKC and subsequently phosphorylate MARCKS, which increases synaptic vesicle availability and glutamate release [33]. Therefore, the inhibitory effect of ISL on Ca^2+^ entry observed in this work may cause a decrease in PKC-induced MARCKS phosphorylation, in turn resulting in decreased glutamate release. Overall, we inferred that ISL, through GABA_B_ receptor activation and the G_βγ_-coupled inhibition of P/Q-type Ca^2+^ channels, suppresses PKC and MARCKS phosphorylation, causing a decrease in evoked glutamate release from rat cerebrocortical nerve terminals (Figure 7).

The concentration (10 μM) of ISL used to depress glutamate release in the present work is consistent with that used in other studies. For example, ISL, at 10 μM, attenuated glutamate-induced increases in intracellular Ca^2+^ and neuronal death in cultured cortical neurons and HT22 hippocampal neuronal cells [9,60]. The inhibition of glutamate release may be a vital mechanism in pathological conditions that cause elevated extracellular glutamate concentrations, such as hypoxia or ischaemia, epileptic seizures, or neurodegeneration [15,61]; several neuroprotective compounds can inhibit glutamate release to ensure extracellular glutamate concentrations remain below neurotoxic levels [17,18,19]. Thus, the inhibition of glutamate release caused by ISL may contribute to explain its anti-excitotoxic actions in vitro [9,10,62] and in vivo [4,5,6,7]. In fact, the beneficial effects of ISL on brain function have been reported following oral and intraperitoneal administration in animals, suggesting that it can cross the blood-brain barrier to reach the brain (Jia et al., 2008; Zhu et al., 2019). However, the brain distribution of ISL needs to be investigated in further studies.

## 5. Conclusions

In conclusion, the inhibitory effect of ISL on glutamate release from cerebrocortical nerve terminals is linked to a decrease in [Ca^2+^]_C_ associated with GABA_B_ receptor activation and the G_βγ_-coupled inhibition of VGCCs as well as the subsequent suppression of PKC and MARCKS phosphorylation. Our finding provided valuable new insights into the mechanism by which ISL operates in the brain and offered a potentially therapeutic treatment for neuronal diseases that involve excessive glutamate release.

## Figures and Tables

**Figure 1 biomolecules-11-01537-f001:**
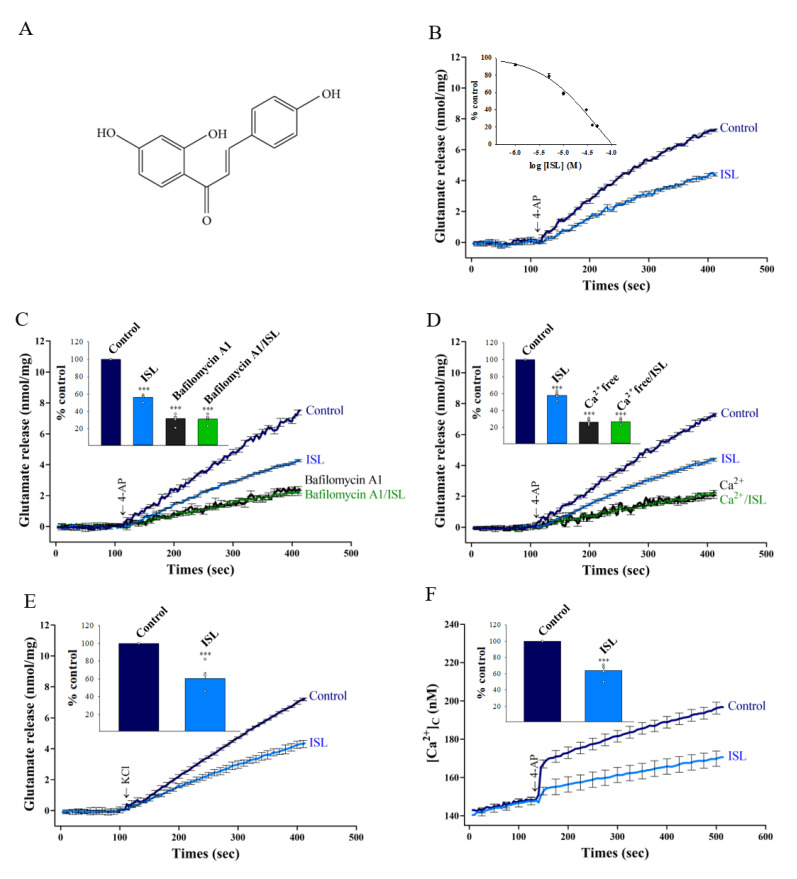
ISL inhibits 4-AP-evoked glutamate release and Ca^2+^ influx from rat cerebrocortical synaptosomes. (**A**) The chemical structure of ISL. (**B**) 4-AP-evoked glutamate release from synaptosomes incubated in the presence of 1.2 mM CaCl_2_, and in the absence (control) or presence of ISL. Inset show dose-response curve of decreases in 4-AP-evoked glutamate release in the presence of ISL (% control release 5 min after 4-AP addition). (**C**) 4-AP-evoked glutamate release from synaptosomes incubated in the presence of 10 µM ISL, 0.1 µM bafilomycin A1, or both (bafilomycin A1/ISL). Inset compares decreases in 4-AP-evoked glutamate release by ISL, bafilomycin A1 or ISL and bafilomycin A1 (% control release 5 min after 4-AP addition). (**D**) 4-AP-evoked glutamate release from synaptosomes incubated in the extracellular Ca^2+^-free solution, and in the presence of 10 µM ISL. Inset compares decreases in 4-AP-evoked glutamate release by ISL, extracellular Ca^2+^-free solution or ISL and extracellular Ca^2+^-free solution (% control release 5 min after 4-AP addition). (**E**) 10 µM ISL-induced inhibition of glutamate release evoked by 15 mM KCl. Inset quantifies the reduction of KC-evoked glutamate release by ISL (% control release 5 min after KCl addition). (**F**) ISL-induced decrease in 4-AP-evoked change in [Ca^2+^]_C_. Inset quantifies the effect of ISL on 4-AP-evoked change in [Ca^2+^]_C_ (% control 5 min after 4-AP addition). Data are the mean ± SEM (n = 5 per group). ***, *p* < 0.001 versus the control group.

**Figure 2 biomolecules-11-01537-f002:**
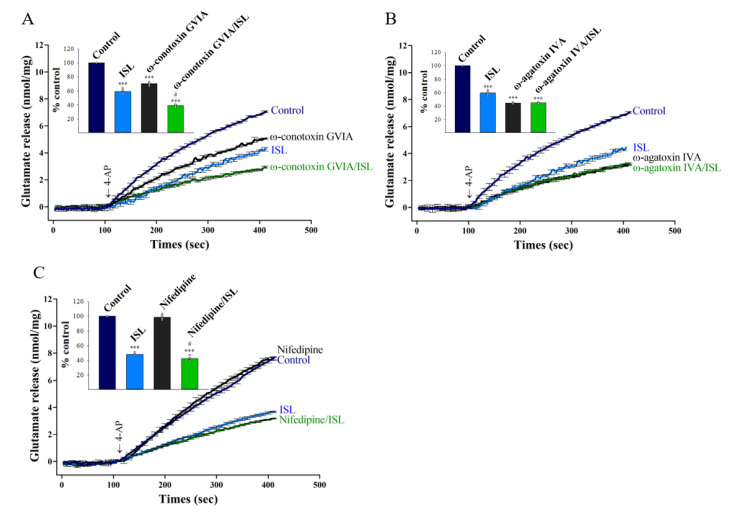
ISL-mediated inhibition of 4-AP-evoked glutamate release in the presence of N-, P/Q-, or L-type VGCC blockade. (**A**) 4-AP-evoked glutamate release from synaptosomes incubated in the presence of 1.2 mM CaCl_2_, and in the absence (control) or presence of 10 µM ISL, 2 µM ω-conotoxin GVIA, or both (**A**); 10 µM ISL, 0.5 µM ω-agatoxin IVA, or both (**B**); or 10 µM ISL, 1 µM nifedipine, or both (**C**). Insets compare the effects of N-, P/Q-, or L-type VGCC blockade on 4-AP-evoked glutamate release, or the inhibition by ISL (% control release 5 min after 4-AP addition). Data are the mean ± SEM (n = 5 per group). ***, *p* < 0.001 versus the control group; #, *p* < 0.05 versus the ω-conotoxin GVIA- or nifedipine-treated group.

**Figure 3 biomolecules-11-01537-f003:**
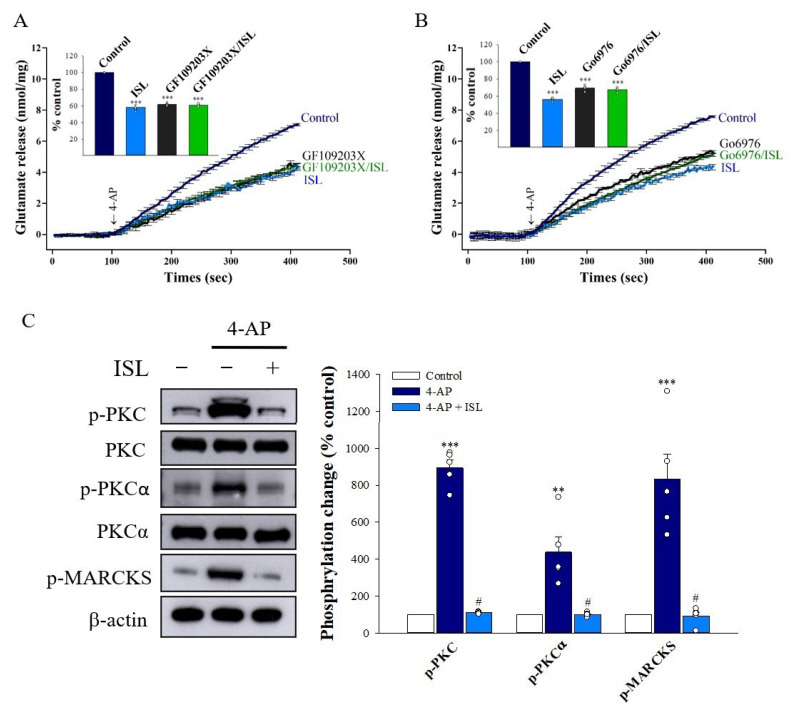
Involvement of PKC suppression in the inhibition caused by ISL on glutamate release. (**A**) 4-AP-evoked glutamate release from synaptosomes incubated in the absence (control) or presence of 10 µM ISL, 10 µM GF109203X, or both (**A**); or 10 µM ISL, 1 µM Go6976, or both (**B**). Insets (**A**,**B**) compare the effects of PKC inhibitors on 4-AP-evoked glutamate release, or the inhibition by ISL (% control release 5 min after 4-AP addition). (**C**) 4-AP-indued phosphorylation of PKC and MARCKS was detected in the absence (−) or presence (+) of 10 µM ISL. The quantification of PKC or MARCKS phosphorylation was normalized to β-actin. Data are the mean ± SEM (n = 5 per group). ***, *p* < 0.001 versus the control group; **, *p* < 0.01 versus the control group; #, *p* < 0.05 versus the 4-AP-treated group.

**Figure 4 biomolecules-11-01537-f004:**
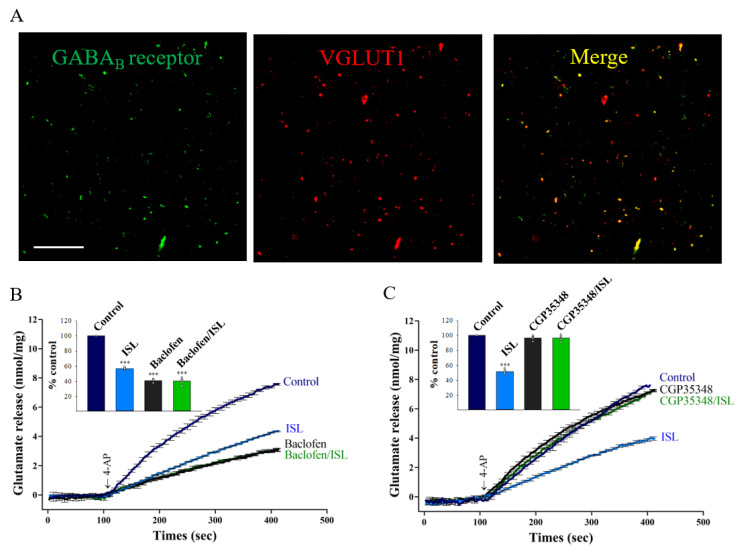
Presynaptic GABA_B_ receptors mediate ISL effects on glutamate release. (**A**) GABA_B_ receptor α1 subunits are present in VGLUT1 positive synaptosomal particles. Confocal microscopy unveiled a significant colocalization of VGLUT1 (red) and GABA_B_ receptor α1 subunit (green) immunopositivities (yellow, merge, arrowhead). Scale bar: 100 μm. 4-AP-evoked glutamate release from synaptosomes incubated in the absence (control) or presence of 10 µM ISL, 50 µM baclofen, or both (**B**); or 10 µM ISL, 100 µM CGP35348, or both (**C**). Insets compare the effects of GABA_B_ receptor agonist or antagonist on 4-AP-evoked glutamate release, or the inhibition by ISL (% control release 5 min after 4-AP addition). Data are the mean ± SEM (n = 5 per group). ***, *p* < 0.001 versus the control group.

**Figure 5 biomolecules-11-01537-f005:**
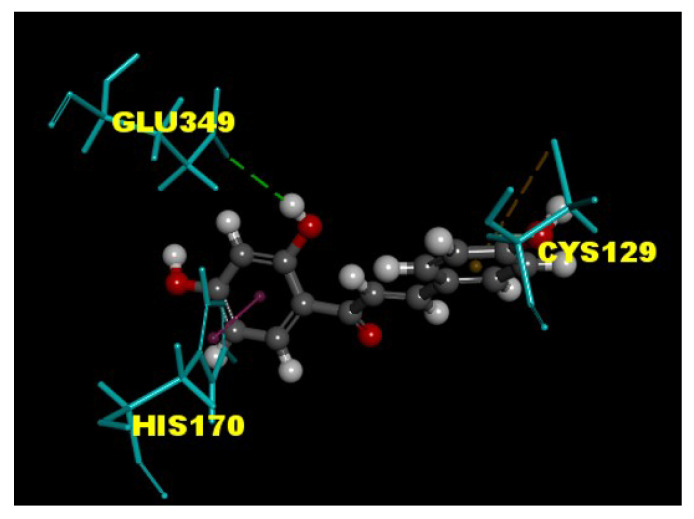
Estimate the binding mode of ISL into GABA_B_ receptor using molecular docking. Binding mode between ISL with the binding site of molecular structure of GABA_B_ receptor (PDB ID 4MGF) was estimated by the Discovery Studio 4.1 software. Protein-ligand hydrogen-bonding and van der Waal interactions are displayed as green and dark purple lines.

**Figure 6 biomolecules-11-01537-f006:**
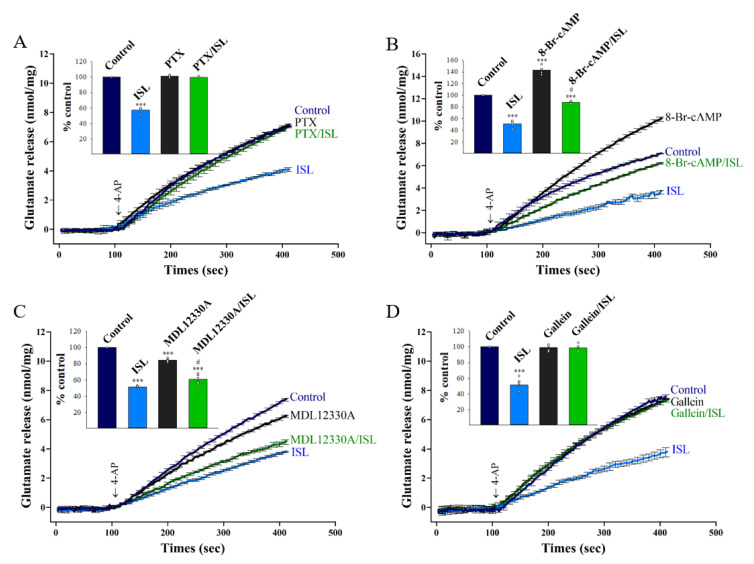
G_βγ_ is involved in ISL-mediated inhibition of glutamate release. 4-AP-evoked glutamate release from synaptosomes incubated in the absence (control) or presence of 10 µM ISL, 2 µg/mL PTX, or both (**A**); or 10 µM ISL, 250 µM 8-bromo-cAMP, or both (**B**); or 10 µM ISL, 10 µM MDL12330A, or both (**C**); or 10 µM ISL, 10 µM gallein, or both (**D**). Insets compare the effects of PTX, 8-bromo-cAMP, MDL12330A, or gallein on 4-AP-evoked glutamate release, or the inhibition by ISL (% control release 5 min after 4-AP addition). Data are the mean ± SEM (n = 5 per group). ***, *p* < 0.001 versus the control group; #, *p* < 0.05 versus the 8-bromo-cAMP- or MDL12330A-treated group.

**Figure 7 biomolecules-11-01537-f007:**
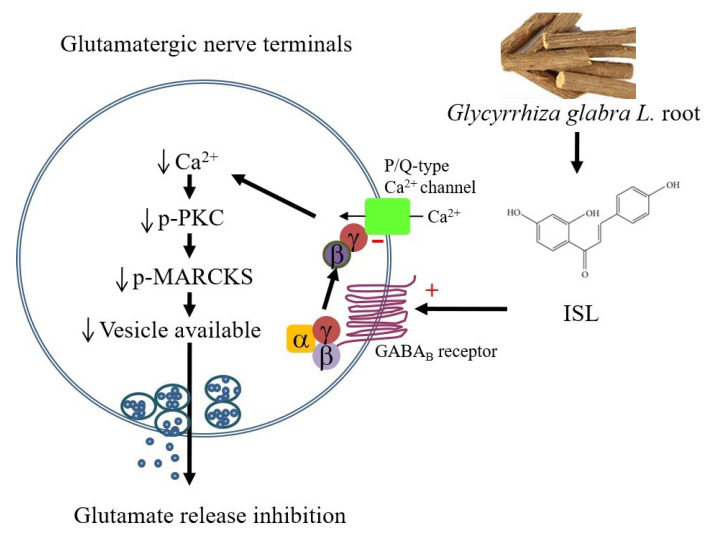
Schematic representation of the main mechanism involved in ISL-mediated inhibition of glutamate release from cerebrocortical nerve terminals. Gi_βγ_, Gi protein βγ subunits; PKC, protein kinase C; MARCKS, myristoylated alanine-rich C kinase substrate.

## Data Availability

The data presented in this study are available on request from the corresponding author.

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
