# Peer review of "Natural Product Isoliquiritigenin Activates GABAB Receptors to Decrease Voltage-Gate Ca2+ Channels and Glutamate Release in Rat Cerebrocortical Nerve Terminals"

_biomolecules, 2021, doi:10.3390/biom11101537_

Round 1

Reviewer 1 Report

In general, the study is based on an experienced experimental protocol already applied in other papers of the group concerning the study of the impact of drugs or natural compounds on glutamate exocytosis. The novelty of the study therefore is limited to the natural compound under study.

The literature regarding ISL must be better described. Furthermore, the union linking neuronal toxicity and glutamate exocytosis is in a way inappropriate since the 4AP-evoked aminoacid release analysed in this study is better consistent with a physiological event rather than a neurotoxic one. In this regard the introduction must be revised in order to avoid useless speculation.

Methods, animals: honestly speaking I would suggest  to reduce the number of animals to that required to obtain reliable results instead of obtaining  consistent effects.

Line 105: five minutes of depolarization, particularly when exposing synaptosomes to hig KCl solution, is a very long period that could affect the efficiency of synaptosomal responses and functional adaptation.

Lines 133-135: synaptosomal lysate and centrifuge and soluble proteins????

Results

Data concerning the impact of ISL concomitantly added to the 4AP stimulus must be introduced in the text.

Line 165: …, indicating the involvement of a decreased exocytotic mechanism….please clarify. In my mind the result is better consistent with the lack of efficacy of ISL in reducing the release of glutamate that does noy originate from the  vesicular store.

Lines 169-170: suggesting the  dependency of the ISL’s action on Ca2+ influx…this is incorrect. The results show that ISL affect the exocytosis of  glutamate which depends on the influx od Ca2+ ions , but does not indicate that the influx of the cation underlies the  ISL’s action.

Figure 1F: the statement “ISL had no significant 176 effect on the basal [Ca2+]C (p = 1; Figure 1F).” please clarify

Line 169: ISL inhibits glutamate release through 196 VGCC suppression.   Please rephrase since the data are best consistent with the conclusion that the ISL-induced inhibition involves the control of the VGCC-mediated events.

Lines 203: please introduce the effect of 10 uM ISL on the 4AP-evoked glutamate release.

The authors must explain why they used 10 µM ISL, since this is a concentration that would be expected to cause a partial inhibition of glutamate exocytosis.

Line 260…”we antagonize….”please control

Lines 270-271: the results are also consistent with the view that ISL and baclofen could share the same mechanism of action

Author Response

Response to reviewer1

biomolecules-1402497R1

We thank the reviewer for the critical comments and constructive suggestions.

Reviewer 1

The literature regarding ISL must be better described. Furthermore, the union linking neuronal toxicity and glutamate exocytosis is in a way inappropriate since the 4AP-evoked aminoacid release analysed in this study is better consistent with a physiological event rather than a neurotoxic one. In this regard the introduction must be revised in order to avoid useless speculation.

As suggestion by the reviewer, several sentences are modified to 〝ISL has received attention because of its various pharmacological benefits, including antibacterial, antiviral, antidiabetic, antioxidant, anti-inflammatory, anticarcinogenic, analgesic, and neuroprotective effects. Lines 43-45); In vitro, ISL attenuates glutamate-, H2O2- or amyloid beta-protein (Aβ) (25‒35)-induced neuronal cell death by reducing the cellular Ca2 + concentration andreactive oxygen species (ROS)levelsin HT22 and PC12 cells, as well as in the cultured cortical neurons of rats. Although the mechanisms underlying the neuroprotective effects of ISL are not fully clarified, it has been reported that this beneficial effect is related to its anti-inflammatory and antioxidant activities (Line 49-54).〞〝Likewise, ISL has a neuroprotective-like effect and whether or not ISLcan influence presynaptic glutamate release should be evaluated (Lines 63-64).

Methods, animals: honestly speaking I would suggest  to reduce the number of animals to that required to obtain reliable results instead of obtaining  consistent effects.

According to this point, the sentence is modified to〝all efforts were made to minimize the number of animals used and their suffering.〞(Line 85)

Line 105: five minutes of depolarization, particularly when exposing synaptosomes to hig KCl solution, is a very long period that could affect the efficiency of synaptosomal responses and functional adaptation.

About this point, the sentence (As synaotosome is not amenable to electrical stimulation, a number of biochemical depolarization protocols have been developed, including the use of K+ channel blocker 4-AP or high external [K+]) is added. (Lines 97-99).

Lines 133-135: synaptosomal lysate and centrifuge and soluble proteins????

According to this point, the sentence is modified to more clear. (Lines 145-146)

Results

Data concerning the impact of ISL concomitantly added to the 4AP stimulus must be introduced in the text.

As suggestion by the reviewer, the sentences 〝4-AP evoked a glutamate release of 7.2 ± 0.1 nmol/mg/5 min. Incubation with 10 µM ISL for 10 min prior to the addition of 1 mM 4-AP produced an inhibition of 4-AP-evoked glutamate release to 4.3 ± 0.1 nmol/mg/5 min〞are added (Llines 171-174).

Line 165: …, indicating the involvement of a decreased exocytotic mechanism….please clarify. In my mind the result is better consistent with the lack of efficacy of ISL in reducing the release of glutamate that does noy originate from the  vesicular store.

As suggestion by the reviewer, the sentence〝indicating the involvement of a decreased exocytotic mechanism〞is deleted (Line 186).

Lines 169-170: suggesting the  dependency of the ISL’s action on Ca2+ influx…this is incorrect. The results show that ISL affect the exocytosis of  glutamate which depends on the influx od Ca2+ ions , but does not indicate that the influx of the cation underlies the  ISL’s action.

As suggestion by the reviewer, the sentence 〝suggesting the  dependency of the ISL’s action on Ca2+ influx〞is deleted (Line 189). In addition, the sentence 〝These results indicate that the observed inhibition of glutamate release by ISL is like to be due to a decrease inCa2+influxthrough VGCCs which are coupled to glutamate exocytosis in the nerve terminals〞is added (Lines 193-196) .

Figure 1F: the statement “ISL had no significant 176 effect on the basal [Ca2+]C (p = 1; Figure 1F).” please clarify

 According to this point, the sentence is modified to 〝10 µM ISL preincubation did not significantly affect basal [Ca2+]C (control, 148.6 ± 1.1; ISL, 147.7 ± 2.1; p = 1 vs. control group; Figure 1F)〞(Lines 197-199)

Line 169: ISL inhibits glutamate release through 196 VGCC suppression.   Please rephrase since the data are best consistent with the conclusion that the ISL-induced inhibition involves the control of the VGCC-mediated events.

As suggestion by the reviewer, the sentence is modified to 〝The results above described indicated that the ISL-induced inhibition of glutamate re-lease involves a reduction in Ca2+ influx through VGCCs〞. (Lines 222-223)

Lines 203: please introduce the effect of 10 uM ISL on the 4AP-evoked glutamate release.

As suggestion by the reviewer, the sentence 〝Specifically, in the absence of GF109203X, ISL significantly reduced the 4-AP-evoked glutamate release〞is added (Lines 258-259)

The authors must explain why they used 10 µM ISL, since this is a concentration that would be expected to cause a partial inhibition of glutamate exocytosis.

As suggestion by the reviewer, the sentences 〝Given the robust repression of evoked glutamate release that was seen with 10 µM ISL, this concentration of ISL was used in subsequent experiments to evaluate the mechanisms that underlie the ability of ISL to reduce glutamate release〞are added (Lines 177-179).

Line 260…”we antagonize….”please control

The word 〝antagonize〞is deleted.

Lines 270-271: the results are also consistent with the view that ISL and baclofen could share the same mechanism of action

 As suggestion by the reviewer, the sentence〝The lack of additivity in the inhibitory actions of ISL and baclofen on glutamate release cans be explained by the inhibition of the same action mechanism by both compounds〞is added (Lines 304-306)

Reviewer 2 Report

In this ms Lin et al studied the function of isoliquiritigenin on glutamate release. The work is interesting and their conclusions are supported by experimental data. Overall, I support the publication of this work, however authors should edit the ms to facilitate the interpretation of the data and of the text. Bellow I suggest some minor changes that can improve the understanding of this ms.  

Minor comments:

Authors should edit some sentences to make them more clear for the readers (either too long or the written English is not very clear). Some examples are lines: 25-28, 157-161, “indicating the involvement of a decreased exocytic mechanism” 165,  260-261, “are was” 263 and 302-306.

“Docking data” in the abstract (line 30) needs some context and/or it needs to be defined. Currently, it is difficult to interpret.

4-AP, line 156, needs to be defined.

In the results section, when the authors describe for the first time the effect of ISL on glutamate release (line 158), it would be nice to have a small description of the essay used. This will make the results more easy to interpret. 

For the statistical analysis authors used one-way ANOVA, which requires normal distributions. Did authors test whether their datasets follow a normal distribution or not? This information should be added to the section about statistical analysis (methods). Moreover, some data sets only have two experimental conditions (e.g. Fig. 1E and Fig. 1F). In these cases, did authors use a t-test or a one-way ANOVA? In the methods section authors only refer to one-way ANOVA, which is not appropriate for data sets with only two experimental conditions. 

In Fig. 2 authors analyzed the effect of ISL on glutamate release in the presence of different inhibitors for different type of VGCCs receptors. It would facilitate the interpretation of the results if authors indicate in the graphs the comparisons between the different inhibitors alone and the inhibitors + ISL. Also, this section lacks a conclusion sentence. What is the main conclusion of this dataset? Does ISL effect on glutamate release need P/Q type receptors? This should be clearly indicated in the text. Actually, this is also true for some of the other sections of results (particularly in section 3.6). The message in the ms will become more clear if authors include a conclusion sentence at the end of each section of results.

Additionally, the interpretation of others graphs (such as Figs 1C, 1D, 3A, 3B, 3C, 4B, 4C and 6A-6D) will also be facilitated if authors indicate the comparisons between the drug/inhibitor and drug/inhibitor + ISL.

CGP35348 is an antagonist of GABAb receptors. This should be clearly indicated in the text (line 265). Currently this is not clear.

The molecular docking section (Fig. 5) lacks some background. Currently, it is very difficult to follow this section in the text. Also, the title of this section (3.5) needs to be tuned down. Authors only showed some modeling, as far as I understand, and there is no experiential data supporting interaction of ISL with GABAb receptors.  

Authors should discuss (either in the results section or in the discussion) their view on how an agonist and antagonist of GABAb receptors can produce the same effect on ISL-mediated inhibition of glutamate release.  

Author Response

Response to reviewer 2

biomolecules-1402497R1

We thank the reviewer for the critical comments and constructive suggestions.

Reviewer 2

Minor comments:

Authors should edit some sentences to make them more clear for the readers (either too long or the written English is not very clear). Some examples are lines: 25-28, 157-161, “indicating the involvement of a decreased exocytic mechanism” 165,  260-261, “are was” 263 and 302-306.

 As suggestion by the reviewer, the sentences are modified (Lines 26-27; Line 185).

“Docking data” in the abstract (line 30) needs some context and/or it needs to be defined. Currently, it is difficult to interpret.

 As suggestion by the reviewer, the sentence is modified to〝docking data revealed that ISL interacted with GABAB receptors 〞(Lines 30-31).

4-AP, line 156, needs to be defined.

 4-AP is defined (Line 99).

In the results section, when the authors describe for the first time the effect of ISL on glutamate release (line 158), it would be nice to have a small description of the essay used. This will make the results more easy to interpret.

 As suggestion by the reviewer, the sentence is modified to 〝To examine the presynaptic action of ISL, isolated synaptosomes were depolarized with 4-APwhich has been shown to open voltage-gated Ca2+ channels (VGCCs) and to induce the release of glutamate〞(Lines 169-171).

For the statistical analysis authors used one-way ANOVA, which requires normal distributions. Did authors test whether their datasets follow a normal distribution or not? This information should be added to the section about statistical analysis (methods). Moreover, some data sets only have two experimental conditions (e.g. Fig. 1E and Fig. 1F). In these cases, did authors use a t-test or a one-way ANOVA? In the methods section authors only refer to one-way ANOVA, which is not appropriate for data sets with only two experimental conditions.

 As suggestion by the reviewer, the sentence is modified to 〝The data were checked for normal distribution and analyzed by Student t-tests or one-way analysis of variance (ANOVA), followed by Fisher’s LSD multiple comparisons tests as appropriate using GraphPad Prism 7 (La Jolla, California). Data are expressed as mean values ± stander error of the mean (SEM) of at least five independent experiments. A p-value < 0.05 was considered statistically significant.〞(Lines 161-165).

In Fig. 2 authors analyzed the effect of ISL on glutamate release in the presence of different inhibitors for different type of VGCCs receptors. It would facilitate the interpretation of the results if authors indicate in the graphs the comparisons between the different inhibitors alone and the inhibitors + ISL. Also, this section lacks a conclusion sentence. What is the main conclusion of this dataset? Does ISL effect on glutamate release need P/Q type receptors? This should be clearly indicated in the text. Actually, this is also true for some of the other sections of results (particularly in section 3.6). The message in the ms will become more clear if authors include a conclusion sentence at the end of each section of results.

 As suggestion by the reviewer, the graphs are modified. Several sentences 〝These results demonstrate that a reduction in Ca2+ influx mediated by P/Q-type VGCCs is associated with the inhibition of glutamate release by ISL (Lines 243-244); These results suggest that the PKC signaling pathway was suppressed by ISL during its inhibition of 4-AP-evoked glutamate release (Lines 276-278); Therefore, these results suggest that GABAB receptor activation is involved in the ISL-mediated inhibition of 4-AP-evoked release (Lines 310-312); These results suggest that ISL-mediated action on glutamate release is dependent on Gβγ subunits, but not on changes in cAMP (Lines 364-365)〞are added.

Additionally, the interpretation of others graphs (such as Figs 1C, 1D, 3A, 3B, 3C, 4B, 4C and 6A-6D) will also be facilitated if authors indicate the comparisons between the drug/inhibitor and drug/inhibitor + ISL.

 As suggestion by the reviewer, the graphs are modified.

CGP35348 is an antagonist of GABAb receptors. This should be clearly indicated in the text (line 265). Currently this is not clear.

 As suggestion by the reviewer, the sentence is modified to〝 Similarly to lack of effect of ISL in the presence of baclofen, ISL had no effect on the 4-AP-evoked glutamate release in the presence of the GABAB receptor antagonist CGP35348〞 (Lines 306-308).

The molecular docking section (Fig. 5) lacks some background. Currently, it is very difficult to follow this section in the text. Also, the title of this section (3.5) needs to be tuned down. Authors only showed some modeling, as far as I understand, and there is no experiential data supporting interaction of ISL with GABAb receptors.

 As suggestion by the reviewer, the sentences〝 In the last decade, machine learning applications is appropriate to use in pharmacological research. Among them, molecular docking is appropriate to use pharmacology to predict the target of natural products (Lines 324-326); Furthermore, the residues His170 and Cys129 in LB1 domain make a van der Waal and a lipophilic interaction with isoliquiritigenin, respectively(Figure 5). Both His170 and Glu 349 are major residues of LB1 domain in GABAB receptor contact to agonists (Lines 332-334)〞are added.

Authors should discuss (either in the results section or in the discussion) their view on how an agonist and antagonist of GABAb receptors can produce the same effect on ISL-mediated inhibition of glutamate release. 

As suggestion by the reviewer, the sentence〝The lack of additivity in the inhibitory actions of ISL and baclofen on glutamate release cans be explained by the inhibition of the same action mechanism by both compounds〞(Lines 304-306) is added .

Reviewer 3 Report

Review Lin

The authors describe the effect of Isoliquiritigenin (ISL) on glutamate release from rat cerebrocortical synaptosomes. The found that ISL inhibits this release via activation of GABAB receptors, which leads to inhibition of P/Q type Ca2+ channels, decrease of the intracellular Ca2+ concentration, and dephosphorylation of PKC and MARKCS.

This is a well-conducted work. The results are well described and convincing and the conclusions are comprehensible.

I have only some proposals to improve the manuscript.

1.) L26: “ISL reduced intraterminal [Ca2+]”
better. “ISL reduced the intraterminal [Ca2+] increase generated by application of 4-AP-induced depolarization”

2.) L29: “ISL was unable to inhibit glutamate release in the presence of baclofen (an GABAB agonist) or CGP3548 (an GABAB antagonist), which was observed through docking data.
This is misleading. The docking data did not demonstrate the interaction of ISL with baclofen or CGP3548.

3.) L103: “…are expressed as nmol/mg/5 min.
Better: “…are expressed as nmol/mg protein/5 min.”

4.) L105: better: FURA-2-AM

5.) L109: “[Ca2+]c (nM) was calculated using the equations described previously
Please provide the equation and the used parameters and describe the calibration protocol.

6.) L123: Please explain the quantification of colocalization.

7.) L160: “the half maximal effective concentration for this inhibition was 17.2 μM
Please discuss this concentration. Will this concentration be reached within the CNS after oral application?

8.) L166: “the glutamate release evoked by 1 mM 4-AP was reduced in the presence of extracellular Ca2+-free solution that contained 300 μM EGTA [26.4% ± 1.5% of control, F(3,16) = 488.7, p < 0.001 vs. control group).
It is not necessary to give numbers if the data are depicted in the graphs. What is the meaning of “F(3,16) = 488.7”?

9.) Several Figures
The ordinate label should be “Glutamate release (nmol/mg)” instead of (nmo/mg).

10.) Fig. 2B: There is still a P/Q-VGCC independent glutamate release. Please, discuss this.

11.) It should be mentioned, that the used method of measuring the continuous glutamate release after 4-AP-induced depolarization of the synaptosomes does not measure the fast SNARE-dependent secretion of the transmitter but rather the late effects, i.e. vesicle docking via Ca2+-CamK-PKC-MARCKS.

12.) L249: “C) 4-AP-indued phosphorylation of PKA and MARCKS was detected in the absence (control) or presence of 10 μM ISL.
Better: C) 4-AP-indued phosphorylation of PKA and MARCKS was detected in the absence (-) or presence (-) of 10 μM ISL.

13.) L250: “Inset quantifies the effect of ISL on 4-AP-evoked phosphorylation …”
This is not an inset. Name it Fig. 3D.

14.) L250: “In these experiments, we antagonist first identified the presence of the GABAB receptor protein”
Better: In these experiments, we identified the presence of the GABAB receptor protein

15.) L271: Omit “The GABAB receptor antagonist CGP35348 yielded a similar result” since this is not a similar result.

16.) Fig. 5: What about the interaction of ISL with Cys129?

17.) L353: “but how ISL affects the P/Q-type Ca2+ channels remains unclear.
No from the presented results it is clear that bg from Gi/o inhibits P/Q type VGCC.

18.) L357: From the cited literature, ISL inhibits K+ channels, which would lead to cell membrane depolarization and opening of VGCC. The inhibition of Na+-channels is probably without effect on the resting membrane potential, since at this potential, most Na+ channels are closed. The membrane potential of the synaptosomes was not measured and therefore, should not be discussed in this way. Most probably, the ISL effect is (according to the presented results) mediated by a direct inhibition of VGCC by the bg -subunits of Gi/o proteins as pointed out below.

19.) The English grammar and style should be improved.

Author Response

Response to reviewer 3

biomolecules-1402497R1

We thank the reviewer for the critical comments and constructive suggestions.

Reviewer 3

1.) L26: “ISL reduced intraterminal [Ca2+]”

better. “ISL reduced the intraterminal [Ca2+] increase generated by application of 4-AP-induced depolarization”

 As suggestion by the reviewer, the sentence is modified to ISL produced a concentration-dependent inhibition of glutamate release and reduced the intra-terminal [Ca2+] increase〞 (Lines 25-27).

2.) L29: “ISL was unable to inhibit glutamate release in the presence of baclofen (an GABAB agonist) or CGP3548 (an GABAB antagonist), which was observed through docking data.”

This is misleading. The docking data did not demonstrate the interaction of ISL with baclofen or CGP3548.

 As suggestion by the reviewer, the sentence is modified to〝docking data revealed that ISL interacted with GABAB receptors 〞(Lines 30-31).

3.) L103: “…are expressed as nmol/mg/5 min.”

Better: “…are expressed as nmol/mg protein/5 min.”

 As suggestion by the reviewer, the sentence is modified to〝nmol/mg protein/5 min 〞(Lines 106, 108).

4.) L105: better: FURA-2-AM

 The word is changed to Fura-2-AM (Line 112).

5.) L109: “[Ca2+]c (nM) was calculated using the equations described previously”

Please provide the equation and the used parameters and describe the calibration protocol.

 As suggestion by the reviewer, the sentences are modified to 〝 Fura-2/Ca fluorescence was monitored in a Perkin-Elmer LS55 spectrofluorimeter at excitation wavelengths of 340 and 380 nm (emission wavelength 505 nm), 340/380 fluorescence ratios (F) were calculated. [Ca2+]C (nM) was calculated following calibra-tion procedures[27], using 0.1% sodium dodecyl sulfate to obtain the maximal fluo-rescence (Fmax) with Fura-2 saturation with Ca2+, followed by 10 mM EGTA (Tris buff-ered) to obtain minimum fluorescence (Fmin) in the absence of any Fura-2/ Ca2+ com-plex.[Ca2+]C (nM) was calculated by the equation[([Ca2+] = Kd (F - Fmin / Fmax – F) [28],using a Kd of 210 nM for the Fura-2/Ca2+ complex〞(Lines 114-121).

6.) L123: Please explain the quantification of colocalization.

 According to this point, the sentence is modified to〝The estimation of the percentage of glutamatergic terminals positive for GABAB receptor was counted three randomly selected areas (255 × 255 µm2) from each coverslip and averaged using ImageJ〞(Lines 133-136).

7.) L160: “the half maximal effective concentration for this inhibition was 17.2 μM”

Please discuss this concentration. Will this concentration be reached within the CNS after oral application?

 According to this point, the sentences〝The concentration (10 μM) of ISL used to depress glutamate release in the present work is consistent with that used in other studies. For example, ISL, at 10 μM, attenuated glutamate‐induced increases in intracellular Ca2+ and neuronal death in cultured cortical neurons and HT22 hippocampal neuronal cells [9,60] (Lines 459-462; In fact, the beneficial effects of ISL on brain function have been reported following oral and intraperitoneal administration in animals, suggesting that it can cross the blood-brain barrier to reach the brain (Jia et al., 2008; Zhu et al., 2019). However, the brain distribution of ISL needs to be investigated in further studies. (Lines 468-472).〞are added.

8.) L166: “the glutamate release evoked by 1 mM 4-AP was reduced in the presence of extracellular Ca2+-free solution that contained 300 μM EGTA [26.4% ± 1.5% of control, F(3,16) = 488.7, p < 0.001 vs. control group).”

It is not necessary to give numbers if the data are depicted in the graphs. What is the meaning of “F(3,16) = 488.7”?

 As suggestion by the reviewer, because the data are depicted in the graphs, the numbers are deleted from the result section.

9.) Several Figures

The ordinate label should be “Glutamate release (nmol/mg)” instead of (nmo/mg).

 The label of Figures is corrected to nmol/mg.

10.) Fig. 2B: There is still a P/Q-VGCC independent glutamate release. Please, discuss this.

 According to this point, the sentences〝In the rat cerebrocortical nerve terminal preparation, the release of glutamate is primarily coupled to the entry of Ca2+ through both N- and P/Q-type Ca2+ channels (39,40) (Lines 220-221); The additive relation between ω-conotoxin GVIA and ISL indicates that N-type VGCCs appear not to mediated the action of ISL on glutamate release. In addition, selective blockade of P/Q-type VGCCs using ω-agatoxin IVA (0.5 µM) reduced control 4-AP-evoked glutamate release (44.7% ± 0.6% of control, p < 0.001). Notably, in the presence of ω-agatoxin IVA, 10 µM ISL inhibition of glutamate release was completely abolished (45.4% ± 0.4% of control, p = 0.92 vs. ω-agatoxin IVA-treated group, Figure 2B), indicating that P/Q-type VGCCs are involved in the observed modulation of glutamate release by ISL. (Lines 230-237)〞are added.

11.) It should be mentioned, that the used method of measuring the continuous glutamate release after 4-AP-induced depolarization of the synaptosomes does not measure the fast SNARE-dependent secretion of the transmitter but rather the late effects, i.e. vesicle docking via Ca2+-CamK-PKC-MARCKS.

 According to this point, the sentences〝Both of these depolarizing treatments are thought to activate VDCCs coupled to glutamate release similarly, and this should be indicated by qualitatively similar modulation if it occurs at the level of the voltage-dependent Ca2+ channel. The two depolarizing paradigms differ in that 4-AP-evoked glutamate release involves the action of Na+ andCa2+ channels, whereas 15 mM external KCl-evoked glutamate release involves only Ca2+channels [30,31]. (Lines 402-407)〞are added.

12.) L249: “C) 4-AP-indued phosphorylation of PKA and MARCKS was detected in the absence (control) or presence of 10 μM ISL.”

Better: C) 4-AP-indued phosphorylation of PKA and MARCKS was detected in the absence (-) or presence (-) of 10 μM ISL.

 The sentence is modified to 〝4-AP-indued phosphorylation of PKC and MARCKS was detected in the absence (‒) or presence (+) of 10 µM ISL.〞(Lines 284-285).

13.) L250: “Inset quantifies the effect of ISL on 4-AP-evoked phosphorylation …”

This is not an inset. Name it Fig. 3D.

 The sentence is modified to〝The quantification of PKC or MARCKS phosphorylation was normalized to b-actin.〞(Lines 285-286).

14.) L250: “In these experiments, we antagonist first identified the presence of the GABAB receptor protein”

Better: In these experiments, we identified the presence of the GABAB receptor protein

 The word〝antagonize〞is deleted.

15.) L271: Omit “The GABAB receptor antagonist CGP35348 yielded a similar result” since this is not a similar result.

 The sentence is modified to〝Similarly to lack of effect of ISL in the presence of baclofen, ISL had no effect on the 4-AP-evoked glutamate release in the presence of the GABAB receptor antagonist CGP35348.〞(Lines 306-308).

16.) Fig. 5: What about the interaction of ISL with Cys129?

 The sentence is modified to〝Furthermore, the residues His170 and Cys129 in LB1 domain make a van der Waal and a lipophilic interaction with isoliquiritigenin, respectively(Figure 5). Both His170 and Glu 349 are major residues of LB1 domain in GABAB receptor contact to agonists [38-40]〞(Lines 332-334).

17.) L353: “but how ISL affects the P/Q-type Ca2+ channels remains unclear.”

No from the presented results it is clear that bg from Gi/o inhibits P/Q type VGCC.

 The sentence is modified to〝Thus, the suppression of Ca2+ influx through P/Q-type Ca2+ channels is involved in the inhibition of glutamate release caused by ISL.〞(Lines 392-394).

18.) L357: From the cited literature, ISL inhibits K+ channels, which would lead to cell membrane depolarization and opening of VGCC. The inhibition of Na+-channels is probably without effect on the resting membrane potential, since at this potential, most Na+ channels are closed. The membrane potential of the synaptosomes was not measured and therefore, should not be discussed in this way. Most probably, the ISL effect is (according to the presented results) mediated by a direct inhibition of VGCC by the bg -subunits of Gi/o proteins as pointed out below.

 In order to make the statement of the sentence more clear, the sentences are modified to 〝Although ISL has been shown to modulate Na+ and K+ channels [3,47,48],the inhibitory effect of ISL on the VGCCs observed in our study is not caused by a change in synaptosomal membrane potential because ISL inhibited the release of glutamate evoked by 4-AP and KCl. Both of these depolarizing treatments are thought to activate VDCCs coupled to glutamate release similarly, and this should be indicated by qualitatively similar modulation if it occurs at the level of the voltage-dependent Ca2+ channel. The two depolarizing paradigms differ in that 4-AP-evoked glutamate release involves the action of Na+ andCa2+ channels, whereas 15 mM external KCl-evoked glutamate release involves only Ca2+channels [30,31]. This indicates that Na+ channels are not involved in the effect of ISL on glutamate release. Furthermore, ISL did not affect the 4-AP-evokedCa2+-independent glutamate release, a component of glutamate release that is exclusively dependent on membrane potential[49,50]. Therefore, our findings indicate that the inhibition of release-coupled VGCCs by ISL reflects a direct effect on VGCC function.〞(Lines 399-412).

19.) The English grammar and style should be improved.

As suggestion by the reviewer, the manuscript is edited by the Wallace Academic Editing.

Round 2

Reviewer 1 Report

the authors introduced the proposed changes and in the revised form it is acceptable for pubblication